# Integrated Bioinformatics Analysis Reveals Marker Genes and Potential Therapeutic Targets for Pulmonary Arterial Hypertension

**DOI:** 10.3390/genes12091339

**Published:** 2021-08-28

**Authors:** Aoqi Li, Jin He, Zhe Zhang, Sibo Jiang, Yun Gao, Yuchun Pan, Huanan Wang, Lenan Zhuang

**Affiliations:** 1Zhejiang Provincial Key Laboratory of Preventive Veterinary Medicine, Department of Veterinary Medicine, College of Animal Sciences, Zhejiang University, Hangzhou 310058, China; 119989@zju.edu.cn (A.L.); jiangsb0219@zju.edu.cn (S.J.); 21717034@zju.edu.cn (Y.G.); 2Department of Animal Science, College of Animal Sciences, Zhejiang University, Hangzhou 310058, China; hejin@zju.edu.cn; 3Institute of Genetics and Reproduction, College of Animal Sciences, Zhejiang University, Hangzhou 310058, China; zhe_zhang@zju.edu.cn (Z.Z.); panyc@zju.edu.cn (Y.P.); 4Key Laboratory of Cardiovascular Intervention and Regenerative Medicine of Zhejiang Province, Department of Cardiology, Sir Run Run Shaw Hospital, Zhejiang University School of Medicine, Hangzhou 310058, China; 5Institute of Preventive Veterinary Medicine, College of Animal Sciences, Zhejiang University, Hangzhou 310058, China

**Keywords:** WGCNA, pulmonary arterial hypertension, hub genes

## Abstract

Pulmonary arterial hypertension (PAH) is a rare cardiovascular disease with very high mortality rate. The currently available therapeutic strategies, which improve symptoms, cannot fundamentally reverse the condition. Thus, new therapeutic strategies need to be established. Our research analyzed three microarray datasets of lung tissues from human PAH samples retrieved from the Gene Expression Omnibus (GEO) database. We combined two datasets for subsequent analyses, with the batch effects removed. In the merged dataset, 542 DEGs were identified and the key module relevant to PAH was selected using WGCNA. GO and KEGG analyses of DEGs and the key module indicated that the pre-ribosome, ribosome biogenesis, centriole, ATPase activity, helicase activity, hypertrophic cardiomyopathy, melanoma, and dilated cardiomyopathy pathways are involved in PAH. With the filtering standard (|MM| > 0.95 and |GS| > 0.90), 70 hub genes were identified. Subsequently, five candidate marker genes (*CDC5L*, *AP3B1*, *ZFYVE16*, *DDX46*, and *PHAX*) in the key module were found through overlapping with the top thirty genes calculated by two different methods in CytoHubb. Two of them (*CDC5L* and *DDX46*) were found to be significantly upregulated both in the merged dataset and the validating dataset in PAH patients. Meanwhile, expression of the selected genes in lung from PAH chicken measured by qRT-PCR and the ROC curve analyses further verified the potential marker genes’ predictive value for PAH. In conclusion, *CDC5L* and *DDX46* may be marker genes and potential therapeutic targets for PAH.

## 1. Introduction

Pulmonary arterial hypertension (PAH) is a rare fatal disease caused by a variety of factors. It is characterized by vascular thickening and vascular obstructive remodeling, which leads to increasing blood pressure in the pulmonary vessels and eventually to heart failure of the right ventricle and even death [1]. According to previous studies, 15 out of every 1 million people have pulmonary hypertension [2]. Current therapies targeting angiotropic and hyperplastic mediators, such as phosphodiesterase-5 (PDE-5) inhibitors, endothelin receptor antagonists (ERAs) and prostacyclin receptor agonists, can result in improved activity endurance and enhanced cardiac function [3,4,5,6]. However, the drugs mentioned above cannot target the other causes of PAH, such as pulmonary vascular remodeling and inflammation, and pulmonary arterial hypertension remains an incurable cardiopulmonary disease [7,8]. Currently, PAH remains incurable. Therefore, elucidating the gene expression profiles of PAH can help to explore the pathogenesis of PAH or seek potential treatments.

The rapidly developing systems biology analyses have become a powerful tool for predicting disease-associated genes, disease subtypes, and disease treatment [9,10]. Weighted gene co-expression network analysis (WGCNA) is broadly employed in investigating microarray samples [11]. Through calculating the correlations between genes, WGCNA could cluster gene collections (modules) with similar expression patterns, analyze the association between modules and sample traits, map the network in each module and identify hub genes [11]. Since it was published, WGCNA has been utilized in numerous fields of biological and medical sciences, such as cancer and mouse genetics [12,13,14].

In this study, we constructed a correlation network with the training dataset. Kyoto Encyclopedia of Genes (KEGG) and Gene Ontology (GO) analyses were also performed to investigate the underlying mechanisms. *CDC5L* and *DDX46* were found in the key module and then assessed by the other dataset and receiver operating characteristic (ROC) curve. Subsequently, the two potential marker genes were double confirmed using quantitative reverse transcription polymerase chain reaction (qRT-PCR) in PAH animal models. Our study may point to potential targets for the treatment of PAH.

## 2. Materials and Methods

### 2.1. Data Selection

The microarray datasets GSE113439, GSE53408, and GSE15197 were downloaded from Gene Expression Omnibus (GEO) (http://www.ncbi.nlm.nih.gov/geo/, accessed on 8 June 2021). GSE113439 includes lung tissues from normal controls (*n* = 11), and PAH patients (*n* = 15); GSE53408 includes lung tissues from normal controls (*n* = 11), and severe PAH patients (*n* = 12); GSE15197 includes lung tissues from normal controls (*n* = 13), idiopathic pulmonary fibrosis patients with secondary pulmonary hypertension (*n* = 8), and PAH patients (*n* = 18). We downloaded the expression data of 18 PAH lung tissues and 13 normal controls from GSE15197 expression profiling for further analysis. Detailed information about the datasets can be found in Appendix A. The workflow for bioinformatics analysis in our study is illustrated (Figure 1).

### 2.2. Data Preprocessing and DEGs Screening

The R software (version 4.0.4) was employed for data analysis. The GSE113439 and GSE53408 datasets in CEL format were annotated according to the Affymetrix Human Gene 1.0 ST Array platform, respectively. The “Affy” package in R [15] was used for handling raw data, including background correction and normalization. GSE15197 was available from Agilent-015420 Whole Human Genome Microarray 4 × 44K G4112F platform. The GSE15197 dataset was downloaded in original txt format. Each dataset was normalized independently with “limma” package in R using RMA [16,17]. 

Probes were switched into gene symbols. Probes not mapped were removed according to the annotation profile of each data set. Several probes annotated to the same gene were collapsed by taking the max expression value of each sample.

The GSE113439 and GSE53408 datasets were merged, and batch effects were then eliminated via the “sva” package [18]. 

According to the workflow, the merged dataset served as a training dataset to identify hub genes, while the GSE15197 dataset was used for assessment.

The effect of inter-sample correction was demonstrated using 3D PCA plots which were performed on the training matrices before and after the removal of inter-batch effect with the “scatterplot3d” [19] package in R. The training dataset was applied for subsequent analyses. DEGs were screened by the “limma” package with the adjusted *p*-value (false discovery rate, FDR) < 0.05 and | (log_2_ FC) | ≥ 1 set as the screening standard. 

### 2.3. GO and KEGG Functional Enrichment Analyses

Functional enrichment analyses were performed using the “clusterProfiler” package [20]. The adjusted *p*-value < 0.05 was considered to be significant.

### 2.4. Co-Expression Network Analysis

The coefficient of variation (CV) of each gene in the training dataset was calculated. Genes with CV > 5% were filtered out and applied as the input for WGCNA.

The co-expression network was generated using the “WGCNA” package (version 1.70-3) in R [11,21]. An appropriate soft threshold ensured that the constructed network fits the scale-free topology (*R*^2^ > 0.85). Hierarchical clustering and dynamic tree cutting functions were used to detect modules (minimum size = 30). The name of each co-expression module was named as the color. 

### 2.5. Identification of Candidate Marker Genes

Through calculating the *p*-value and Pearson’s correlation coefficient of module eigengenes (MEs) and the disease trait of each module, we could identify the key module which is most relevant to PAH.

Module membership (MM) stood for the correlation between MEs and the profile of gene expression. The gene significance (GS) of genes in the module was then calculated, representing the correlation between genes and disease traits. Genes possessing high MM and GS in the key module were highly interrelated with the disease trait.

With the key module selected, we defined |MM| > 0.95 and |GS| > 0.90 as the screening criteria for filtering hub genes in the key module.

To further target more candidate marker genes, the chosen module’s corresponding gene relation matrices were imported into the Cytoscape plugin, CytoHubba [22], which helped to identify the top 30 hub genes through two different calculation methods, namely EcCentricity and BottleNeck. Finally, the overlapped five candidate marker genes identified by the three methods above were obtained.

### 2.6. Validation of Candidate Marker Genes and ROC Curve Analyses 

Expressions of candidate marker genes within the training and validating datasets were illustrated for assessment. ROC curve analyses were performed by the “pROC” package [23].

### 2.7. PAH Model and qRT-PCR

The cDNA samples of control and PAH lung tissues were provided kindly by Dr. Xun Tan’s Laboratory of Zhejiang University as gifts. PAH was induced in broiler chickens by cool temperature exposure. An indicator of right ventricular hypertrophy, the ratio of the right ventricular weight to the whole ventricular weight (RV/WV) was measured. Chicken with RV/WV > 0.28 were diagnosed as PAH [24].Then, each 96-well plate well was mixed with 6.5 μL of cDNA (diluted at 1:50), 7.5 μL of qPCR SYBR Mix (Tsingke, Hangzhou, China), 1 μL of primers. The primers used in the study are listed in Appendix A.

### 2.8. Statistical Analysis

Expression of hub genes between the two groups performed unpaired *t*-tests using GraphPad Prism (version 9.0). *p*-value < 0.05 was assigned as significance.

## 3. Results

### 3.1. Data Preprocessing and DEGs Screening

The GSE113439 and GSE53408 expression matrices were combined, containing lung samples from 27 controls and 22 PAH patients. Next, the batch effects were eliminated and the sample distributions before and after adjustment for batch effects were illustrated in the 3D PCA plots (Figure 2A,B).

A total of 542 DEGs were identified. A volcano plot and DEG heatmaps are illustrated in Figure 2C,D. The expression of the DEGs also indicated the differences between the control and PAH groups.

### 3.2. GO and KEGG Analyses of DEGs

In the GO analyses, the most enriched biological process (BP) terms were associated with ribosome biogenesis, regulation of mitotic nuclear division, and mitotic cytokinesis. The most enriched terms for cellular components (CC) were mainly associated with pre-ribosome and centriole. The most enriched molecular function (MF) terms were associated with DNA-dependent ATPase activity and DNA helicase activity (Figure 3A–C). 

In the KEGG analysis, the DEGs were enriched in ribosome biogenesis in eukaryotes, melanoma, hypertrophic cardiomyopathy, and dilated cardiomyopathy (Figure 3D).

### 3.3. Construction of WGCNA Network and Identification of the Key Module

After filtering genes with CV > 5% in the training dataset, 3926 genes were filtered and applied as input for WGCNA network construction. Through Pearson’s correlation analysis, no outlier samples were found (Figure 4A). 

Next, the power of β = 5 determined by the function “sft$powerEstimate” was selected so as to construct a scale-free network, which is similar to the real state of biological network, because it corresponded a power-law distribution (Figure 4B,C). Eight modules were identified in the gene expression net according to the dynamic shearing method. A dendrogram of the hierarchical clustering of genes and the assigned module colors are shown in Figure 4D. 

Our study only focused on the relationship of PAH with each module. Therefore, we only calculated the correlation between MEs and PAH measured by the *p*-value and Pearson’s correlation coefficient. The results show that turquoise module (cor = 0.91, *p*-value = 8 × 10^−20^) was most significantly correlated with PAH. Additionally, the calculation of connectivity versus gene significance for each module, further verified the turquoise module as the key module correlated with PAH (Figure 5A,B).

To further demonstrate the correlation between the PAH and turquoise module, the values between MM versus GS in each module were computed. The turquoise module (cor = 0.97, *p*-value < 10^−200^) showed a strong correlation with PAH (Figure 5C). The value of MM versus intramodular connectivity (IC) double confirmed the turquoise as a high correlation with PAH (cor = 0.92, *p*-value < 10^−200^) (Figure 5D). These results indicate that genes in the turquoise module tend to highly correlated with PAH.

### 3.4. GO and KEGG Analyses of the Key Module

On the basis of the results above, the turquoise module was selected as the key module for subsequent analyses. GO analysis of the turquoise module showed that it is mainly enriched in ribosome biogenesis, pre-ribosome, centriole, DNA helicase activity, DNA-dependent ATPase activity, and tRNA binding (Figure 6A–C).

KEGG analysis was also performed on the key module. The significant enrichments were mainly related to arrhythmogenic ventricular cardiomyopathy, melanoma, hypertrophic cardiomyopathy, and dilated cardiomyopathy terms (Figure 6D).

### 3.5. Identification of Hub Genes

According to the truncation standard (|MM| > 0.95 and |GS| > 0.90), 70 hub genes were identified out. In addition, we imported the top 500 genes in the key module into Cytoscape. The PPI network between imported genes was constructed using the Cytoscape software. CytoHubba, a Cytoscape plugin, was used to glean the top 30 hub genes of biological network analysis with Bottle Neck and eCCentricity algorithm. Subsequently, the overlapping candidate marker genes selected by the three methods mentioned above were used to identify potential marker genes, and they were illustrated as a Venn diagram using the R package “Venn Diagram”. The overlapped five candidate marker genes: *CDC5L*, *AP3B1*, *ZFYVE16*, *DDX46*, and *PHAX* may play important role in PAH (Figure 7A).

### 3.6. Validation of Candidate Marker Genes

In the training dataset, the overlapped five candidate marker genes were up-regulated in the PAH samples significantly (*p*-value < 0.001). In the validating dataset GSE15197, we found that the expressions of *CDC5L* and *DDX46* were significantly up-regulated in the PAH samples. (*p*-value < 0.01). (Figure 7B).

We observed that broiler chickens also suffered from PAH and could serve as excellent PAH models for exploring the mechanism of PAH. The cDNA samples were extracted from lung tissues of an independent set of control and PAH chickens. The qRT-PCR results obtained from the cDNA samples indicate that both *Cdc**5l* and *Ddx46* were up-regulated significantly in the PAH group (*p*-value < 0.001) (Figure 7C).

### 3.7. ROC Curve Analyses of Hub Genes

ROC curve analyses were performed to verify the potential marker genes, and area under the curve (AUC) values were calculated. In the training dataset, *CDC5L* (AUC = 0.997), *DDX46* (AUC = 1), *ZFYVE16* (AUC = 1), *PHAX* (AUC = 1), *AP3B1* (AUC = 1) showed excellent predictive powers (Figure 8A). The verification results of GSE15197 dataset further substantiate the fact that *CDC5L* (AUC = 0.782), *DDX46* (AUC = 0.842), *ZFYVE16* (AUC = 0.765), *PHAX* (AUC = 0.568), *AP3B1* (AUC = 0.769) showed excellent predictive power (Figure 8B).

## 4. Discussion

PAH is characterized by loss of the pulmonary vascular and obstructive remodeling, causing disability and premature death [1]. Drug therapies have improved symptoms and signs of PAH, but are unable to cure the disease fundamentally. Thus, novel markers and potential targets are needed to seek. Our study merged two datasets for predictive analysis, and identified the key module highly correlated with PAH via WGCNA. After obtaining hub genes by means of the correlation between GS and MM, overlapping the hub genes with top thirty genes calculated by two different calculation methods in CytoHubb, we discovered two hub genes, *CDC5L* and *DDX46*, which were further validated in the GSE15197 dataset and animal experiments. Our findings may help to find new marker genes and potential targets for currently uncurable PAH.

Due to different criteria regarding sample selection, experimental conditions, and microarray platforms between different datasets, the gene expression levels always show variations in different datasets. In the training dataset, we identified the overlapped five candidate marker genes, but not all the candidate marker genes could be verified by the validating dataset GSE15197. To remove the variations, we expanded the sample size by merging two datasets for training. There are three genes, *ZFYVE16*, *PHAX*, and *AP3B1*, that were not validated by another microarray dataset, while the remaining two potential marker genes, *CDC5L* and *DDX46*, were further verified by qRT-PCR. To test the specificity of the candidate genes in PAH subtypes, expression analyses were performed on the GSE113439 dataset, one component of the training dataset with classification of PAH subtypes. As shown in Appendix A, *CDC5L*, *DDX46*, *AP3B1*, and *ZFYVE16* were significantly dysregulated in several subtypes of PAH. *PHAX* was not specific in CTEPH. *ZFYVE16* was reported to be associated with CTEPH [25], which is consistent with our result.

Although not validated by the GSE15197 dataset, the three candidate marker genes may be involved in PAH. Belonging to the FYVE structural domain family, the candidate marker gene *ZFYVE16* participates in pathways closely related to PAH, such as the TGF-Beta, BMP and EGFR signaling pathways [26,27]. A recent study detected its sequence variants in peripheral blood of CTEPH patients [25]. Additionally, it has been identified as one of the candidate genes related with PAH [28]. *AP3B1* was reported to lead to Hermansky–Pudlak syndrome type 2 both in humans and mice, associated with pulmonary fibrosis [29,30]. Meanwhile, the role of AP3B1 in PAH remains unknown, and it needs to be further explored in the future. PHAX (adaptor for RNA export), essential for the transportation of small nucleolar RNA (snoRNA) and small nuclear RNA (snRNA), was reported to be a marker of PAH [31,32]. A recent study showed that PHAX is required for DNA damage response, which is deeply associated with PAH [33,34].

Potential marker gene *CDC5L* codes a DNA-binding protein involved in cell cycle regulation, and complexing with PRP19 participates in response to DNA damage [35]. It has been implicated in bladder cancer, prostate cancer, and hepatocellular carcinoma tumorigenesis [36,37,38]. According to the Mouse Genomics database, *Cdc5l*^−/−^ mice led to preweaning lethality or embryonic lethality. Previous studies identified *CDC5L* as a bridge gene linking lung adenocarcinoma and chronic obstructive pulmonary disease [39]. DNA damage could be observed in the nucleus and mitochondria in PAH patients. The relationship between DNA damage and PAH has been widely discussed, and the most popular types of DNA damage and their causes in PAH remain to be completely investigated [34]. In addition, a very recent study also identified *CDC5L* as a marker gene in PAH using comparative transcriptional analysis [40]. However, the function of *CDC5L* in PAH still remains elusive and needs to be further explored.

*DDX46*, also known as *PRP5*, is one of the components of the 17S U2 snRNP (small nucleus ribonucleoprotein) complex belonging to the DEAD-box family of RNA helicases. A previous study showed that *Ddx46* null mice tend to demonstrate embryonic or preweaning lethality [41]. In vitro biochemical experiments showed that *DDX46* protein is required for stable association of U2 snRNP and spliceosome, and serves several functions before and through pre-spliceosome assembly in pre-mRNA splicing [42,43]. The abnormal expression of *DDX46* has also been reported in tumors [44,45,46]. The expression of *DDX46* is increased in esophageal squamous cell carcinoma and colon cancer tissues, promoting cell growth and inhibiting apoptosis [44,46,47]. *DDX46* is abnormally expressed in osteosarcoma tissues and cells, promoting cell growth and invasion [48]. Interestingly, a study previously reported that DDX46 interacts with SMC4, which was identified substantially involved in lung development and highly related with the progression of lung adenocarcinoma [49]. Therefore, by interacting with SMC4, DDX46 may contribute to the development of the lung and may be associated with PAH by means of the GSK-3β/β-catenin pathway [50,51,52].

The pathophysiological mechanism of PAH is complex, and its development is influenced by environmental and genetic factors. Pathogenic mutations in several genes, such as *BMPR2* and *ACVRL1*, have been reported to be associated with PAH [53,54,55]. Of the known pathogenic mutant genes, *BMPR2* has been observed in approximately 29% of PAH patients [56]. In our study, although genetic tests were not performed on the analyzed samples, the expression of *BMPR2* (adjusted *p*-value = 0.031591, log_2_FC = 0.196004) showed significant difference to a mild extent (Appendix A). The TGF-β pathway has been proven to have an important role in the pathogenic of PAH [57]. In the merged dataset, expression of *SMAD2* (adjusted *p*-value = 6.34 × 10^−10^, log_2_FC = 0.51) and *TGFBR2* (adjusted *p*-value = 0.039, log_2_FC = 0.16) differed in PAH (Appendix A), although to a lesser extent. These results indicate that both BMPR2 and TGF-β pathway indeed altered the PAH pathogenicity 

In the training dataset, we identified 542 DEGs and found the turquoise module with the strongest correlation with PAH. The GO and KEGG annotation results of DEGs and the turquoise module demonstrate that the pre-ribosome, ribosome biogenesis, centriole, DNA dependent ATPase activity, DNA helicase activity, hypertrophic cardiomyopathy, melanoma, and dilated cardiomyopathy pathways were all enriched, which is in line with current research findings [58,59,60,61,62,63,64].

Previously named ascites syndrome in broilers, PAH occurs in approximately 3–5% of broilers when raised at a low temperature. Hence, fast-growing broiler chickens could serve as an excellent model for PAH. The main pathogenic factor to PAH in broiler chickens is assumed to be an autonomic insufficiency in vascular capacity of the lungs [65,66]. Right ventricular hypertrophy, considered as the indicator for judging broiler pulmonary hypertension, is an intuitive physiological manifestation of pulmonary hypertension. Through qRT-PCR, we found that the two potential marker genes, *CDC5L* and *DDX46*, were significantly upregulated, which shows that the genes are not human-specific.

As a result of limited samples and a lack of detailed sample information, our study has some limitations: (1) the accuracy of disease assessment and prediction can be improved if the sample size is enlarged and genetic information is completed; (2) the potential marker genes and pathways identified in this study need to be further verified to provide solid evidence for clinically targeted therapies; (3) analyses of protein expression levels of the candidate genes could provide more evidence on the potential implications of marker genes in PAH

## 5. Conclusions

In summary, by integrating and analyzing multiple datasets, and through the establishment of a co-expression network via WGCNA analysis, two potential marker genes of PAH, *CDC5L* and *DDX46*, were identified. Through GO and KEGG analyses, the ribosome biogenesis, centriole, ATPase activity, helicase activity, hypertrophic cardiomyopathy, melanoma, and dilated cardiomyopathy pathways may relate to PAH. Together, our findings provide new perspectives for the understanding of the pathogenesis of PAH.

## Figures and Tables

**Figure 1 genes-12-01339-f001:**
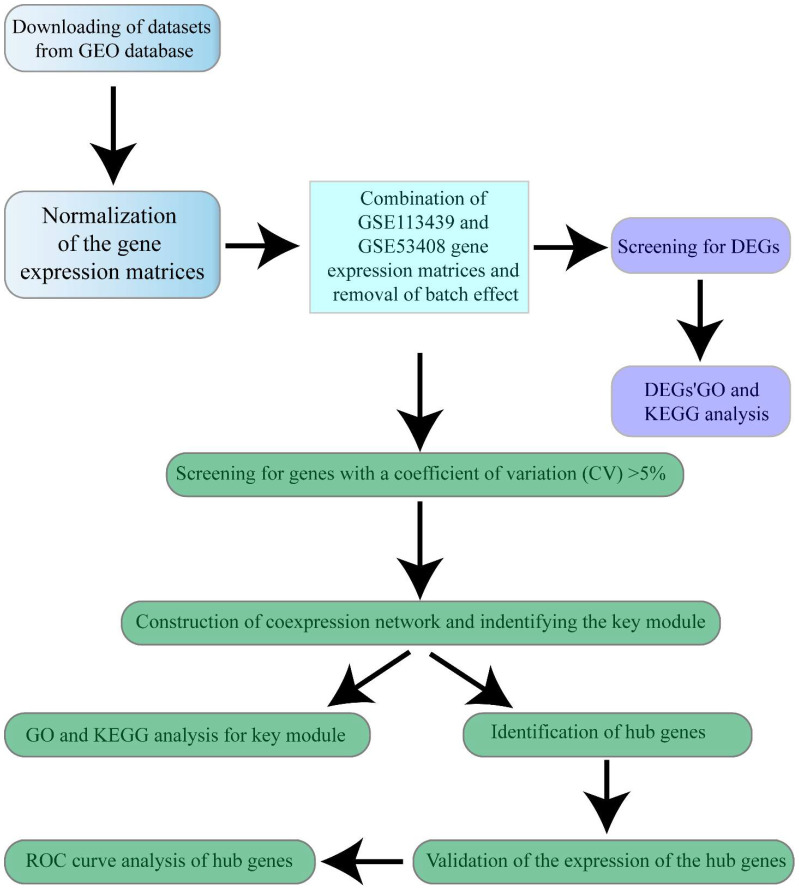
Workflow of the study design.

**Figure 2 genes-12-01339-f002:**
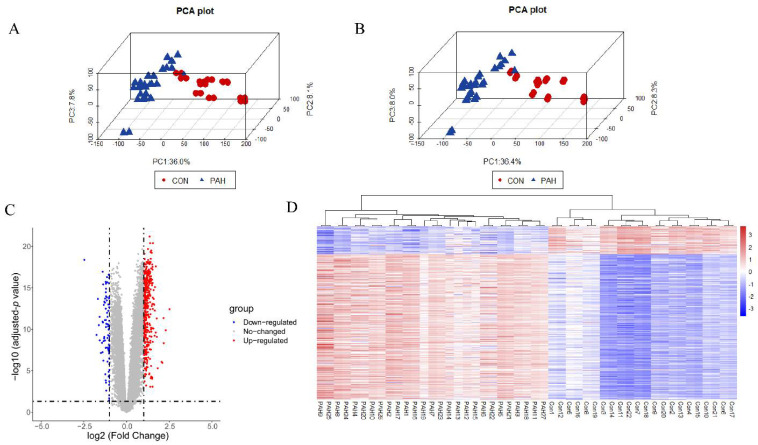
Datasets merging and DEG screening. (**A**,**B**) Three-dimensional PCA plots before and after the removal of inter-batch effect. (**C**) Volcano plot. (**D**) Heatmap of all DEGs. CON, normal lung tissue; PAH, lung tissues of pulmonary arterial hypertension patients; DEGs, differentially expressed genes.

**Figure 3 genes-12-01339-f003:**
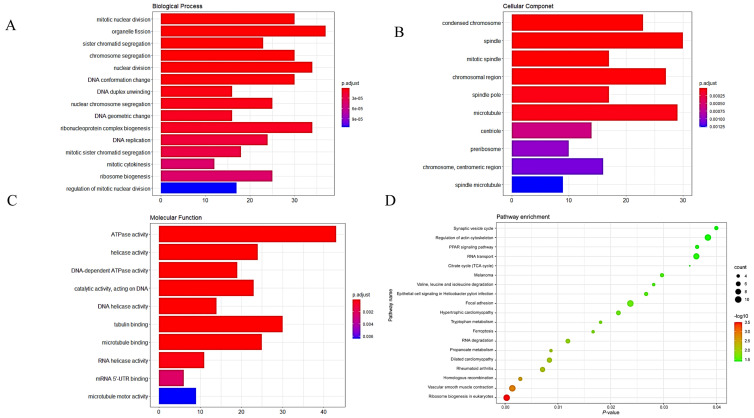
GO and KEGG analyses of DEGs. (**A**–**C**) GO analyses of DEGs. (**D**) KEGG analysis of DEGs. BP, biological process; CC, cellular component; MF, molecular function.

**Figure 4 genes-12-01339-f004:**
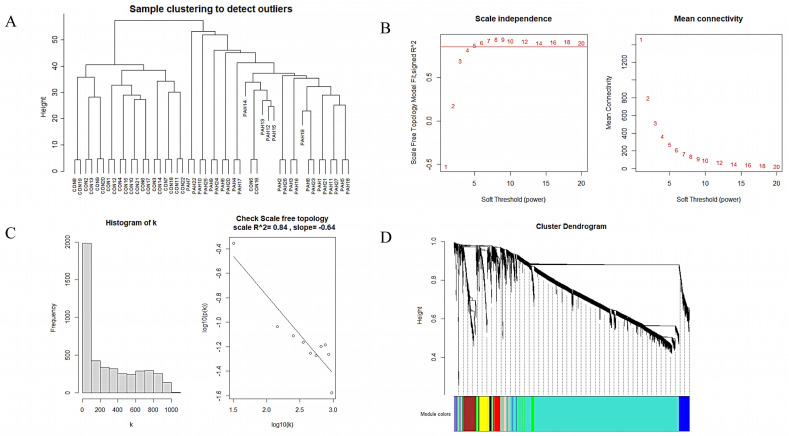
Construction of WGCNA network. (**A**) Detection of outliers. (**B**) Suitable soft threshold powers selection, the red line indicating 0.85. (**C**) Examining the scale free topology when β = 5. (**D**) The cluster dendrogram of genes in the WGCNA network. CON, normal lung tissue; PAH, lung tissues of Pulmonary arterial hypertension patients.

**Figure 5 genes-12-01339-f005:**
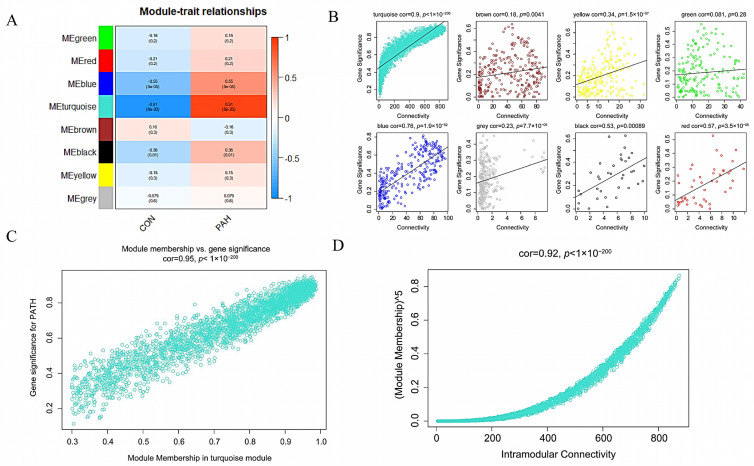
Identification of the key module. (**A**) Correlations between the gene module and PAH. (**B**) Scatterplot of connectivity versus GS for PAH of each module. (**C**) Scatterplot of GS for PAH versus MM in the turquois module. (**D**) Scatterplot of IC versus MM in the turquois module. MM, module membership; GS, gene significance; IC, intramodular connectivity.

**Figure 6 genes-12-01339-f006:**
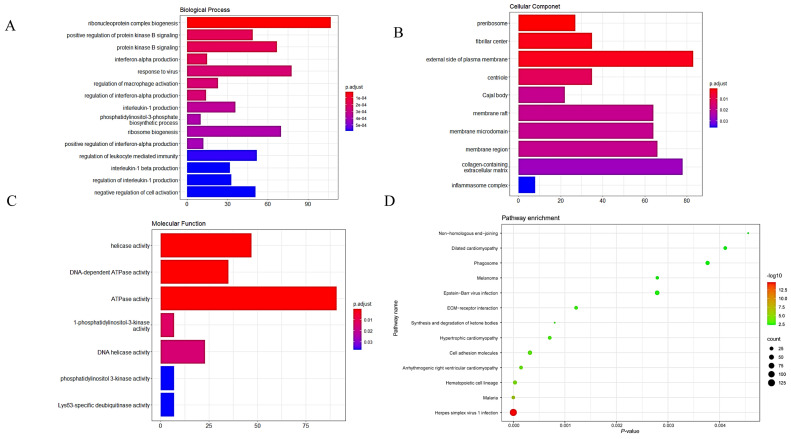
GO and KEGG analyses of the key module. (**A**–**C**) GO analysis of the key module. (**D**) KEGG analysis of the key module. BP, biological process; CC, cellular component; MF, molecular function.

**Figure 7 genes-12-01339-f007:**
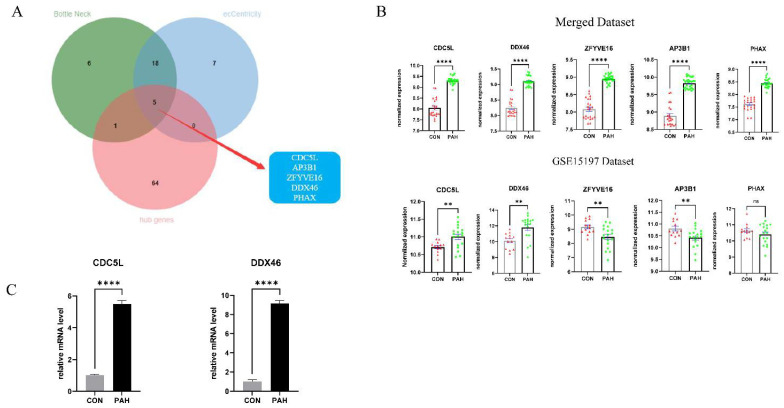
Identification and validation of the candidate marker genes. (**A**) Overlapped candidate marker genes calculated by three different calculation methods. Namely, examining GS and MM, top thirty genes calculated by two different calculation methods in CytoHubb. (**B**) Expression of candidate marker genes in the training and the validating datasets. (**C**) Expression of potential marker genes in chicken PAH.; ** *p*-value < 0.01; **** *p*-value < 0.0001; CON, normal lung tissue; PAH, lung tissues of pulmonary arterial hypertension.

**Figure 8 genes-12-01339-f008:**
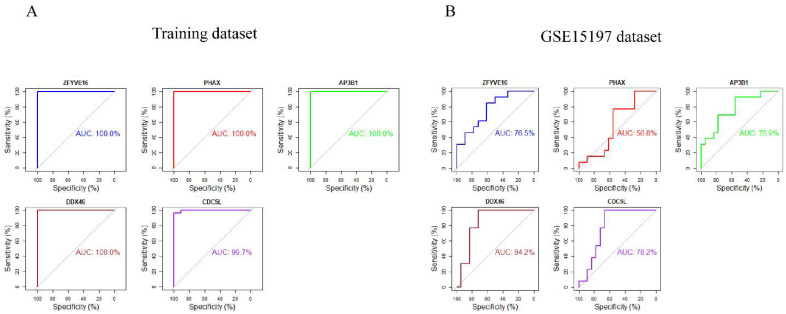
Validation of five candidate marker genes by ROC curve analyses. (**A**) ROC curve analysis of *ZFYVE16*, *PHAX*, *AP3B1*, *DDX46* and *CDC5L* in the training dataset. (**B**) ROC curve analysis of *ZFYVE16*, *PHAX*, *AP3B1*, *DDX46* and *CDC5L* in the validating dataset GSE15197.

## Data Availability

The data presented in this study are available upon request from the corresponding author.

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
