# Peer review of "Integrated Bioinformatics Analysis Reveals Marker Genes and Potential Therapeutic Targets for Pulmonary Arterial Hypertension"

_genes, 2021, doi:10.3390/genes12091339_

Round 1
Reviewer 1 Report
Overall a nice application of systems biology approaches to pulmonary arterial hypertension. I had a few questions/comments when reading the manuscript.
- There are many places in the manuscript where it is said that CDC5L and DDX46 are involved in "development of PAH" or “genes of the turquoise module play an important role in PAH" - I think this language needs to be softened, since it is not clear that these genes are truly involved in the pathogenesis and the association could be related to downstream effects of PAH.
- A few times the need for "biomarkers" for PAH is mentioned - while not exactly specified, the authors seem to imply biomarkers for PAH diagnosis (given ROC analysis that was performed for PAH vs control). I'm not sure of the clinical significance for biomarkers for diagnosis of PAH vs controls given that clinically these groups of patients are very different
- The "turquoise" module is referred to often in the abstract, intro, and results. It would be helpful in the methods to clarify what this means
- Are the causes of PAH in available in the datasets?
- It is mentioned that the modules were "correlated with clinical characteristics" - what clinical characteristics were these?
- There is a mention of the identifying "potential prognostic genes" in the manuscript - but gene expression was not actually correlated with any prognostic features
- In the Discussion I don’t agree with the phrase “Drug therapies have improved symptoms and signs of PAH but not mortality” – IV prostacyclin therapy has improved mortality in randomized clinical trials
- It would be helpful to discuss whether CDC5L or DDX46 has been identified in prior work evaluating PAH lung (or PBMC) gene expression
- I wasn't sure why the lack of difference in BMPR2 expression was discussed, as it wasn't clear that any of the samples carried the germline BMPR2 variant
Reviewer 2 Report
The manuscript is a well written and well explained about the DEGs in lung tissue samples from patients with PAH.
I have some major comment regarding the study design and some other considerations:
- All patients from which lung tissue sample was extracted belong to the same PAH group according to the Nice 2018 classification?
- In all patient samples, genetic test was performed in order to know the possible genetic background of these samples? This is extremely important, as a genetic mutation can alter the pattern of gene expression and therefore it could introduce a bias in the analysis.
- Germinal mutations have been also test in all samples for the top five genes detected by the group? CDC5L, AP3B1, ZFYVE16, DDX46, and PHAX?
- How the authors explain that the difference in expression in the top five genes was different among the merge dataset and the GSE15197 dataset? i.e. ZFYVE16 has overexpression in the merge dataset but under expression in the GSE15197 dataset? The same thing occurs with AP3B1 and PHAX.
- Have the authors analyzed the phenotype of possible mouse models in which there are a knock-out / knock down of any of the candidate genes they proposed? MGI has the largest collection of mouse models and this can be useful to be checked
- Due to the fact that TGF-Beta pathway in one of the main actor in the PAH pathogenicity biological model, have the authors detected any differential in expression in genes within this pathway compare to controls?
Minor comments
- Page 12 lines 308 to 313 this paragraph belong to the material and method and results
Round 2
Reviewer 1 Report
Appreciate the revisions and the responses
Reviewer 2 Report
Many thanks for the authors for the revision of the manuscript, it has been improved based on the comments by the reviewers. However, there is still a lack of information regarding some important points of the study results and conclusion:
- What is the role and utility of the detected candidate genes with different expression? Are this genes specific for any subtype of PAH or could be dysregulated in several subtypes?
- What the authors could suggest to confirm or discard the potential implications of these genes in the pathogenisis of PAH?
- The manuscript need a final version without track changes
- There are some minor typos that has to be reviewed by the authors